# Quantitative Lithiation Depth Profiling in Silicon Containing Anodes Investigated by Ion Beam Analysis

**Sören Möller** [1,*] , **Hyunsang Joo** [2] , **Marcin Rasinski** [1] , **Markus Mann** [1] , **Egbert Figgemeier** [2,3,4] and **Martin Finsterbusch** [1]

1. Forschungszentrum Jülich GmbH, Institut für Energie und Klimaforschung, 52425 Jülich, Germany; m.rasinski@fz-juelich.de (M.R.); m.mann@fz-juelich.de (M.M.); m.finsterbusch@fz-juelich.de (M.F.)
2. Helmholtz Institute Münster (HI MS), IEK-12, Forschungszentrum Jülich GmbH, Jägerstr. 17–19, 52066 Aachen, Germany; h.joo@fz-juelich.de (H.J.); e.figgemeier@fz-juelich.de (E.F.)
3. Institute for Power Electronics and Electrical Drives (ISEA), RWTH Aachen University, Jägerstr. 17–19, 52066 Aachen, Germany
4. Jülich Aachen Research Alliance, JARA-Energy, 52425 Jülich, Germany
* Correspondence: s.moeller@fz-juelich.de

**Abstract:** The localisation and quantitative analysis of lithium (Li) in battery materials, components, and full cells are scientifically highly relevant, yet challenging tasks. The methodical developments of MeV ion beam analysis (IBA) presented here open up new possibilities for simultaneous elemental quantification and localisation of light and heavy elements in Li and other batteries. It describes the technical prerequisites and limitations of using IBA to analyse and solve current challenges with the example of Li-ion and solid-state battery-related research and development. Here, nuclear reaction analysis and Rutherford backscattering spectrometry can provide spatial resolutions down to 70 nm and 1% accuracy. To demonstrate the new insights to be gained by IBA, $SiO_x$-containing graphite anodes are lithiated to six states-of-charge (SoC) between 0–50%. The quantitative Li depth profiling of the anodes shows a linear increase of the Li concentration with SoC and a match of injected and detected Li-ions. This unambiguously proofs the electrochemical activity of Si. Already at 50% SoC, we derive $C/Li = 5.4$ ($< LiC_6$) when neglecting Si, proving a relevant uptake of Li by the 8 atom % Si ($C/Si \approx 9$) in the anode with $Li/Si \leq 1.8$ in this case. Extrapolations to full lithiation show a maximum of $Li/Si = 1.04 \pm 0.05$. The analysis reveals all element concentrations are constant over the anode thickness of 44 μm, except for a ~6-μm-thick separator-side surface layer. Here, the Li and Si concentrations are a factor 1.23 higher compared to the bulk for all SoC, indicating preferential Li binding to $SiO_x$. These insights are so far not accessible with conventional analysis methods and are a first important step towards in-depth knowledge of quantitative Li distributions on the component level and a further application of IBA in the battery community.

**Keywords:** all-solid-state batteries; garnet solid-state electrolyte; lithium-ion transport; electrochemical stability; ion-beam analysis; depth-resolved analysis

## 1. Introduction

Si has stayed in the limelight as a promising material to supersede graphitic carbon, the currently predominant element of the negative electrode, in commercial lithium-ion batteries (LIB). The almost factor 10 higher specific capacity and nearly three times higher volumetric capacity as well as stability, natural abundancy, and a proper operating potential range (~0.2–0.4 V vs. $Li/Li^+$) make this material stand out [1–3]. Nonetheless, Si has rarely been applied to commercial LIBs, especially with regard to the application for an electric vehicle, due to several fatal drawbacks such as a 200–300% volume change while undergoing lithiation and delithiation [4,5], a large amount of Li-loss during the first cycle [6,7], and a high ionic and electrical resistance, which in turn boosts the cell degradation and exacerbates the rate capability. A number of strategies have been proposed

to alleviate these defects in a wide variety of ways: constructing a nanocomposite [8–11], coating Si particles with a robust material [12,13], and developing a new binder [14–16] or electrolyte additives [17–20]. Even though they increased the feasibility of applying Si to the commercialized cells, a Si-based anode has yet to outstrip the current state-of-the-art anode active material, graphite, from a practical point of view. Introducing silicon oxides ($SiO_x$) has been regarded as one of the viable alternatives to utilizing pure Si, since the oxides generated during cycling ($Li_2O$, $Li_4SiO_x$, etc.) act as buffer material for significantly reducing the volume change [21–26]. As a result, $SiO_x$ has more stable cyclability than pure Si, despite its lower specific capacity [1,24,27]. On the other hand, due to the additional reactions forming lithium silicates, the lithiation reactions in $SiO_x$ are more complex, hindering further understanding of driven improvements. This work, therefore, studies an anode with $SiO_x$ added to a graphite anode, which is a highly feasible material but still needs to be understood throughout its lithiation process. Since Si starts to be reduced at a higher potential range in comparison with graphite [28] and decomposition of the electrolyte, and its additives is dominated at the beginning of lithiation [29], it is particularly valuable to demonstrate the underlying phenomena at low state-of-charge (SoC) levels. Hence, this work focuses on the concentration gradients of the composition of the anode at early lithiation stages to comprehend the actual lithiation degree of Si and graphite.

The investigation of lithium distribution during (dis-)charging battery cells especially along the depth direction remains a difficult task since it requires a method sensitive to the light element while providing 1% range accuracy, sub-µm spatial resolution, and a fast time resolution required by fast charging and discharging processes. The 3D nature of the Li migration and the layered cell structure potentially induce non-constant depth profiles and lateral variations due to inhomogeneous migration rates, electrical resistances, and fundamental limits. In battery cells, this can limit the battery performance or even lead to fatal failure mechanisms such as dendrite formation [30]. Consequently, the knowledge of the migration dynamics enables revealing fatigue processes and bottlenecks within the cell stack. A deeper understanding of the local SoC and state-of-health (SoH) is desired for the development of new cell types (e.g., all-solid state cells), commercial purposes, and further performance improvements.

Numerous techniques exist, fulfilling these tasks to certain extents. Discussions of various methods can be found in reviews, e.g., [31]. However, most of the widely used methods enable only to analyse the distribution of the electrode composition on a specific surface level or in a limited volume range. For example, even though data from X-ray photoelectron spectroscopy (XPS) contains chemical bondings as well as their relative quantity, this information mostly comes from the outermost atomic layer of the sample. The distribution can be conveniently captured by energy-dispersive X-ray spectroscopy (EDX), but it only allows a small range with limited depth resolution and has a lower accuracy especially for the light atoms, in particular, lithium. Inductively coupled plasma–optical emission spectroscopy (ICP-OES) allows for quantifying the sample composition, but requires a step for dissolving the sample into solution, which precludes spatially-resolved measurement [32]. One of the recent strategies to investigate the content of electrode composition along the depth direction is to utilize glow discharge optical emission spectroscopy (GD-OES). This method analyses spectroscopic data from excited atoms released by argon plasma sputtering. It provides an elemental depth profile in a porous electrode within a micrometer-scale depth range, but suffers from the matrix effect, requiring complementary methods or complex calibration steps [32,33]. Neutron depth profiling (NDP), as a nuclear technique, offers superior depth resolution, detection limits, and direct quantification [34]. Its non-destructive nature and the negligible heat load induced by the thermal neutrons allow for operando analysis of cells. Its analytical range remains limited, though, since the neutrons can penetrate deeply, but the emitted ions have a limited range in the order of µm. The count rates limit the time-resolution to about 5–15 min per point. NDP works best with thick and large cells and only detects specific isotopes, namely $^6$Li for lithium-based batteries, limiting its capability to determine full stoichiometries.

Ion-beam analysis (IBA) offers a possible solution for the other methods limitations or at least provides complementary information. IBA has the fundamental advantage of providing the complete compositional depth-resolved information together with the lithium information, enabling to reveal the full stoichiometry of every cell component in a single measurement. With this, its application is in no way limited to Li-based batteries, but this work will focus on Li batteries as the most common example. NDP and GD-OES offer a better depth resolution (no projectile straggling), but IBA offers the better lateral resolution (focussing of the beam with spot sizes down to ~1 µm) compared to NDP and GD-OES. The IBA information depth depends on the projectile and products, allowing for a higher degree of flexibility in terms of range and spatial resolution compared to optical methods or NDP by proper selection of projectile ion species and energy. Therefore, IBA potentially provides valuable information on the cell dynamics, degradation processes, and manufacturing aspects as an input to advanced cell modelling, research and development, and quality assurance.

Several challenges are connected with IBA of batteries, due to the requirements of low beam-induced heat-loads in sensitive battery materials, a conflict with high counting statistics, accuracy, and small beam spots. Furthermore, limited analysis ranges and the requirement of vacuum impose practical drawbacks, in particular when compared to photon-based methods. Careful optimisation of the analysis setup enables a feasible solution in this optimisation space, even allowing for operando electrical connections of those cells [35].

In the literature, IBA of cathodes [36,37], electrolytes [38], and full cell assemblies with µm-resolution of its components ex-situ and operando was proven quite recently by a few groups [39–41]. This work first goes a step back and investigates the fundamental possibilities of several IBA methods and parameters suitable for conducting such an analysis. Different reactions and projectiles are compared to find the optimal solution for lithium battery-specific questions. Nuclear reaction analysis (NRA) and Rutherford backscattering spectrometry (RBS) are then applied ex-situ to a 20 weight% $SiO_x$-doped graphite anode material charged in a coin-cell set-up to six different SoC from 0–50% to investigate its lithium depth profiles and other compositional aspects. Data analysis methodology and uncertainties are presented in order to conclude on future prospects of the method in the battery field.

## 2. IBA Method in View of Lithium Analysis

Battery materials typically consist of mostly light elements (Li, C, O), a few 10% of intermediate elements (Si, Ni, Co, Ti . . . ), and sometimes minimal amounts of heavy elements (La, Ta . . . ). The latter two can be analysed using particle-induced *x*-ray emission analysis (PIXE) and RBS with theoretically available cross-sections and a single known reaction (RBS) or peak group (PIXE). The light element analysis requires NRA or particle-induced gamma-ray emission analysis (PIGE). NRA and PIGE feature several reaction options for each light element and require measured cross-sections for each of them. Consequently, the analysis of batteries requires combining at least two IBA methods and selecting the right reactions for the light elements.

For IBA of lithium, several nuclear reactions are possible, each with individual range, resolution, and detection limits, see Table 1. Only depth probing by penetration is considered, as the depth resolution for side analysis of cross-cuts only depends on ion beam focussing and spot size and is typically worse compared to the depth resolution. The depth resolutions for the individual reactions are analysed using the software RESOLNRA 1.7 from the SimNRA7 (Garching, Germany) package [42]. The analysis employed 3 MeV projectiles at perpendicular incidence and a reaction angle of 165° with a 15 keV FWHM detector resolution, if not stated differently, probing into pure $LiCoO_2$. $LiCoO_2$ is taken as an example material for Li-based cell materials since it features a representative Li concentration and stopping power for many materials. An ion beam only sees passed atoms, not passed distance. This makes IBA insensitive to porosity or crystallographic volume changes

e.g., upon dis-/charging or phase transitions. Therefore, a standard $LiCoO_2$ density of 5.05 g/cm$^3$ corresponding to $1.25 \times 10^{23}$ atoms/m$^2$ µm can be used for recalculating the depth in units of atoms/m$^2$ to the geometric depth or the other way round for any given mass density of the sample.

Table 1 compares the possible reactions with H, D, and $^3$He reactions for Li analysis by NRA. The $^7$Li(p,$\alpha_0$)$^4$He reaction offers the best compromise between resolution, range, and practical aspects. The $^7$Li(d,$\alpha_0$)$^5$He has similar properties but so far lacks in cross-section data, and the decay of $^5$He potentially leads to increased background levels and lower accuracy. The $^6$Li(p, $^3$He)$^4$He reaction has the best depth resolution, but due to the low Q-value resulting in low energy products, an overlap with the RBS reactions of heavier elements such as Co and their pile-up strongly limits its practical range. For following the isotopic ratios in $^6$Li-enriched materials, it might be suitable due to higher signal levels, though. Below 2200 keV proton energy, a window without overlap of the $^6$Li(p, $^3$He)$^4$He peak with typical battery elements such as Mn exists, which was used to test the existing cross-sections for this reaction using thin films. The Chia-Shou Lin data [43] for 147.1° appear to match the experimental spectra at 150°, but they offer only a few data points. The Bashkin data [44] offer more data points, but were found to be largely incorrect, at least for 150°. The data suggest a better sensitivity due to larger cross-sections at higher reaction angles. The application of deuterium ions and beam energies >3 MeV potentially provides an improved range or depth resolution, but an application is so far limited due to the lack of reaction cross-sections. All three considered ions provide suitable reactions for the separate analysis of $^6$Li and $^7$Li and hence could be applied to enrichment-based lithium migration studies. Reactions with $^3$He projectiles have significantly lower range, but similar depth resolution as H and D induced reactions. In contrast to the Li resolution, the detection and resolution for other elements is 4x better with $^3$He compared to H and D.

In addition to these reactions, all ions can also excite the first nuclear level of $^7$Li(p, pg$_{1-0}$)$^7$Li at 477.6 keV for PIGE analysis. A relevant PIGE cross-section is given for >1 MeV protons with a mostly monotonous increase towards higher energies (no resonances). The working of PIGE enables ranges from $280 \times 10^{22}$ at./m$^2$ for 2 up to about $6000 \times 10^{22}$ at./m$^2$ at 10 MeV proton energy (~0.5 mm in $LiCoO_2$). The 477.6 keV photon energy results in negligible photon absorption (~1%) even for the thickest samples. A depth resolution can be generated by using several beam energies and generating differences. The depth resolution then depends on the ratio of beam energy width to stopping power at a certain depth. This value varies from ~3 to ~$600 \times 10^{22}$ at./m$^2$ in the range stated above, resulting in generally much lower depth resolution for Li compared to NRA.

For most of the Li reactions discussed here, especially the $^7$Li(p,$\alpha_0$)$^4$He reaction, the product energy is hardly influenced by the reaction angle or projectile energy. Hence, a narrow aperture according to the setup described in [35] is chosen to reduce the detectors angular acceptance, leading to a better depth resolution (geometrical straggling). In general, larger reaction angles (towards 180°) lead to improved depth resolution, but are technically more challenging due to the detector size. $^7$Li(p,$\alpha_0$)$^4$He can achieve a surface near resolution of $1.7 \times 10^{22}$ at/m$^2$ = 136 nm in $LiCoO_2$ in our standard setup, hence the analysed cell systems can be thin-film or bulk cells. The maximum probing depth at 3 MeV is ~22 µm, but below 12 µm the signal will interfere with the RBS part of the heavier constituents. This situation is mostly independent of the beam energy.

**Table 1.** Comparison of the possible NRA reactions of lithium with 3 MeV light ions. For recalculation to length use e.g., the dense $LiCoO_2$ density of 0.08 $\mu m/(10^{22} at/m^2)$ corresponding to 5 $g/cm^3$ or a porous Si/graphite anode as used later of 0.164 $\mu m/(10^{22} at/m^2)$.

| Reaction | $^7Li(p,\alpha_0)^4He$ | $^7Li(d,\alpha_0)^5He$ | $^7Li(^3He,\alpha_0)^6Li$ | $^7Li(d,p_0)^8Li$ | $^7Li(^3He,p_0)^9Be$ | $^6Li(^3He,p_0)^8Be$ | $^6Li(p,\,^3He)^4He$ | $^6Li(^3He,\,\alpha_0)^5Li$ | $^6Li(d,\,\alpha_0)^4He$ |
|---|---|---|---|---|---|---|---|---|---|
| Q-value [MeV] | 17.34 | 14.23 | 13.33 | −0.19 | 11.2 | 16.79 | 4.02 | 14.91 | 22.37 |
| Range [$10^{22}$ at/$m^2$] | 280 | 230 | 55 | 110 | 70 | 50 | 50 | 50 | 350 |
| Resolution [$10^{22}$ at/$m^2$] | 1.70 | 1.70 | 2.15 | 2.30 | 3.50 | 7.70 | 0.65 | 2.60 | 2.75 |
| Cross-section | Paneta [45] | n.a. >1.6 MeV | Not available | Paul [46] | Not available | Not available | Chia-Shou Lin [43] | Not available | Only <2 MeV |

Figure 1 demonstrates the scaling of depth resolution with the analysis setup properties and the depth inside $LiCoO_2$. The reaction products can be detected to a depth of $310 \times 10^{22}$ at./m², but the RBS edges begin to overlap with the NRA reaction in this range with ~100 times higher intensity, limiting the final range. Depending on the setup and the elements present in the material, pulse pile-up can further reduce the effective range. In fact, ~3 MeV protons provide the highest analysis depth range, since for lower energies the cross-section strictly decreases; while for higher energies, the elastic scattering edges shift to higher energies while the $^7Li(p,\alpha_0)^4He$ product energy remains mostly constant. Only methodical additions such as PIGE would allow further extending this range. For analysing Li in the Si-doped carbon anode material investigated later, the range, in terms of atoms/m², is within 10% of the values depicted above. The range in terms of μm is significantly higher, depending on the material density. For a density of the anode material used below of 1.4 g/cm³, we expect a Li probing range of ~60 μm. Generally, we expect similar ranges and resolutions in units of atoms/m² for all typical lithium battery materials due to the similar relevant elemental ranges and compositions.

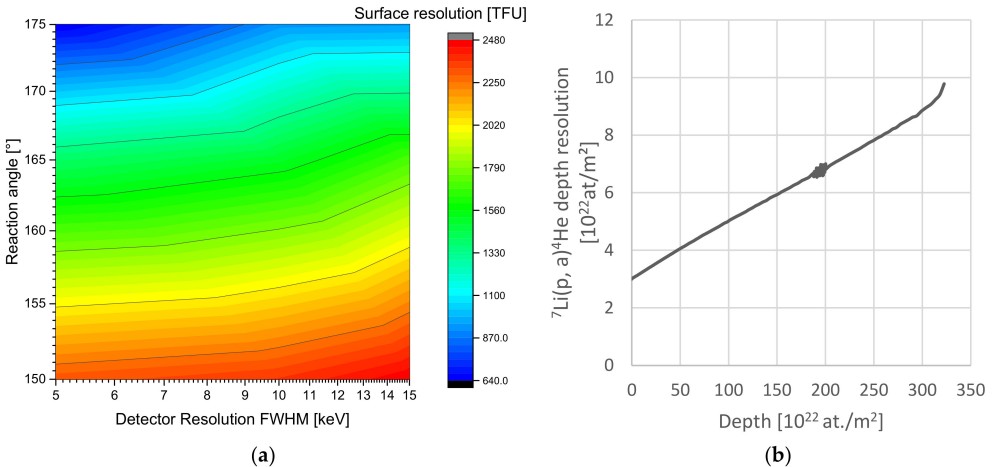

**Figure 1.** (**a**) Near-surface depth resolution in [TFU = $10^{19}$ at/m²] vs. detector resolution and angle for normal beam incidence of the $^7Li(p,\alpha_0)^4He$ reaction. The impact of resolution is minor compared to the angle. (**b**) Resolnra calculation of the depth resolution in the Si-doped anodes showing the increase of depth resolution with depth in our given setup of 15 keV energy resolution and 150° reaction angle. The depth resolution roughly triples up to the analysis range of $3 \times 10^{24}$ at./m² equal to ~30 μm with a linear increase.

In conclusion, for the analysis of lithium, the $^7Li(p,\alpha_0)^4He$ reaction provides the best compromise in terms of resolution, range, and practical properties. It requires high reaction angles, as close as possible to 180°, in order to optimize depth resolution and sensitivity. Typical commercially available silicon-based detector energy resolutions of 10 keV FWHM suffice for the analysis with only little gain expected from improved energy resolution. On top of that, every (dis-)charged Ah corresponds to moving a certain number of lithium nuclei, and every lithium nucleus corresponds to a certain (small) number of counts in the spectrum, further limiting the detection properties statistically (minimum resolved Ah step). The μNRA setup [35] provides up to 200 nm depth resolution, which is sufficient considering the limits of counting statistics, although ~70 nm is technically possible with a technically optimised setup employing a maximum reaction angle and energy resolution.

For assessing the radiation damage in the material, the SRIM 2013 code (*Quick Calculation of Damage* mode, $10^5$ particles) is applied according to [47,48]. We use the output "Total Target Vacancies", which equals an integral over the depth, as the displacements per incident ion (DPI) [49]. As the exact values for the displacement threshold are not known, the standard SRIM value of 28 eV for carbon is used in a representative material of 64% C and 9% H, O, Si, and Li each. The ions penetrate up to 95 μm, assuming a density of 1.83 g/cm³, inducing

23.3 DPI. If we restrict the calculation to a depth closer to our actual material thickness in this study of 50 μm, the calculation yields a mostly constant 4.2 DPI throughout the target depth. For comparison, the typical active material $LiCoO_2$ with an assumed displacement threshold of 25 eV yields relatively constant 7.9 DPI up to 35 μm depth with a total range of 47 μm and 26 DPI. Calculating the integral damage during analysis with 10 μC on 3.14 mm$^2$ spot area yields a total damage of $1.3 \times 10^{-5}$ displacements per atom. $1.5 \times 10^{-4}$ of the H projectiles remain in the sample, coming to a total of $3 \times 10^{15}$ H/m$^2$. These small numbers seems negligible compared to the typical defect densities and H impurities. In conclusion, neither displacement damage nor H implantation can significantly influence the composition or structure of battery materials during our analysis.

The upper limit of ion beam-induced heating can be estimated by equating Planck's radiation law and the beam power as described in [50]. This conservative approach assumes zero heat conduction, but only losses by thermal radiation. An ion-beam energy of $E_{ion}$ = 2960 keV, a beam current $I_{Beam}$ = 5 nA, an emissivity $\varepsilon$ = 1 (black surface), and a beam spot area of $A_{Beam}$ = 3.14 mm$^2$ yield a maximum sample temperature of 364 K. Finite element simulations show the conduction in the copper substrate reduces the sample temperature increase during analysis to a few K above room temperature. In conclusion, the ion beam-induced temperature increase could influence the ionic and electronic conductivity of the cell in extreme cases during in-situ analysis, but it will not induce chemical changes or reactions, influencing the typical post-mortem analysis.

## 3. Sample Preparation

### 3.1. Material and Electrode Preparation

The anode slurry contains 20 wt.% of silicon-oxide $SiO_x$ (AS2D, ShanShan Tech, Shanghai, China); 70 wt.% of graphite (SMG-A5, Hitachi, Japan); 2 wt.% of carbon black (C65, Imerys, France); and 8 wt.% of binders composed of carboxymethyl cellulose (CMC 2000PA, DOW Chemical, Midland, USA), styrene-butadiene rubber (SBR) (BM-451B, Zeon Europe GmbH, Düsseldorf, Germany), and lithium substituted polyacrylic acid (LiPAA). Graphite and $SiO_x$ have similar densities of 2.24 and 2.26 g/cm$^3$, but different particle sizes of 11–23 and 5–7 μm, respectively. The slurry is then coated on a copper foil (10 μm thickness), dried at 60 °C, and subsequently calendered. The density of the produced anode is 1.375 g/cm$^3$ or 0.164 μm/(10$^{22}$ at./m$^2$). The anode is punched to 14 mm diameter using a high-precision electrode cutter (EL-CELL) and then dried for longer than 10 h at 80 °C in a glass oven B-585 Drying (Buchi). The dried anode is transferred to an argon-filled glove box to be assembled with a lithium foil (PI-KEM, Wilnecote, UK) in a coin cell set-up (2032-type). A glass microfiber filter (GF/C, Whatman, UK) is used as a separator and 1.0 M $LiPF_6$ in ethylene carbonate (EC): ethyl methyl carbonate (EMC) (3:7, *w/w*) (E-Lyte Innovations GmbH, Münster, Germany) mixed with 10 wt.% of fluoroethylene carbonate (FEC) (E-Lyte, Germany) as an electrolyte. Figure 2 demonstrates the cell assembly.

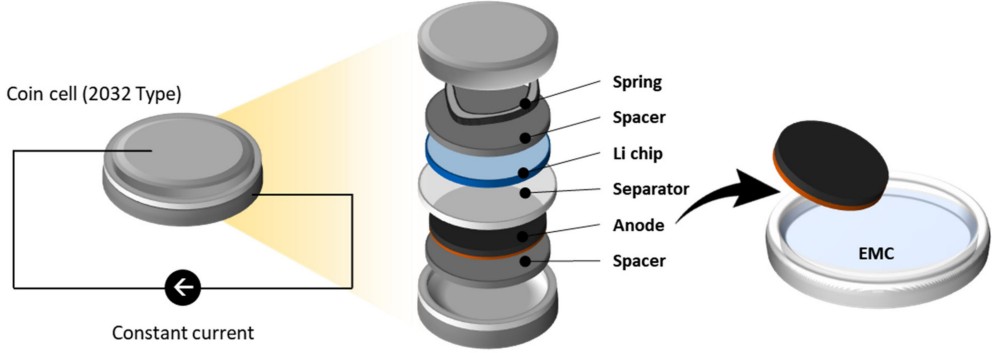

**Figure 2.** Schematic diagram of sample preparation. In this study, the part named "anode" is analysed for its elemental composition. The coin cell set-up is used for the lithiation process. The anode is extracted from the set-up and rinsed with EMC.

### 3.2. Lithiation Process

The assembled coin cells are lithiated with a galvanostatic mode at 0.2 C-rate until the lithiation capacity of the cells reaches the designed degrees as shown in Figure 3a. Table 2 describes the lithiation parameters for the different SoCs. However, prior to this process, the full capacity of the anode is measured with a cut-off potential of 0.01 V vs. Li/Li$^+$ to 622 mAh/g, which is defined as the 100% SoC in this study. This value agrees to a calculation using 340 mAh/g for C and 1700 mAh/g for SiO$_x$ together with the deposited masses. The open circuit voltage (OCV) of each cell is monitored for 10 h after lithiation to check the potential recovery, as shown in Figure 3b, which may reflect lithium migration inside the structure. When the increasing trend of OCV subsides, the cells are carefully disassembled to take the lithiated anodes out. The anodes are rinsed with EMC for 5 min and dried under vacuum at room temperature.

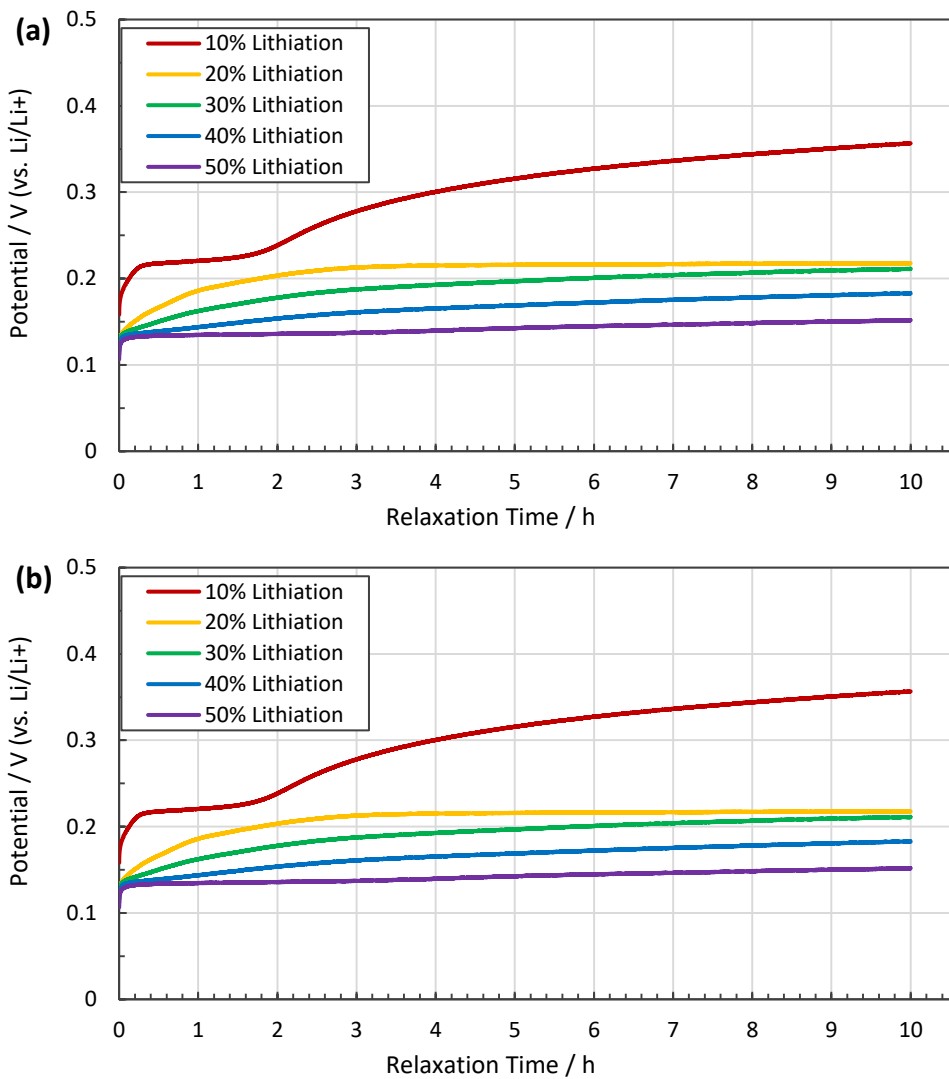

**Figure 3.** *Cont.*

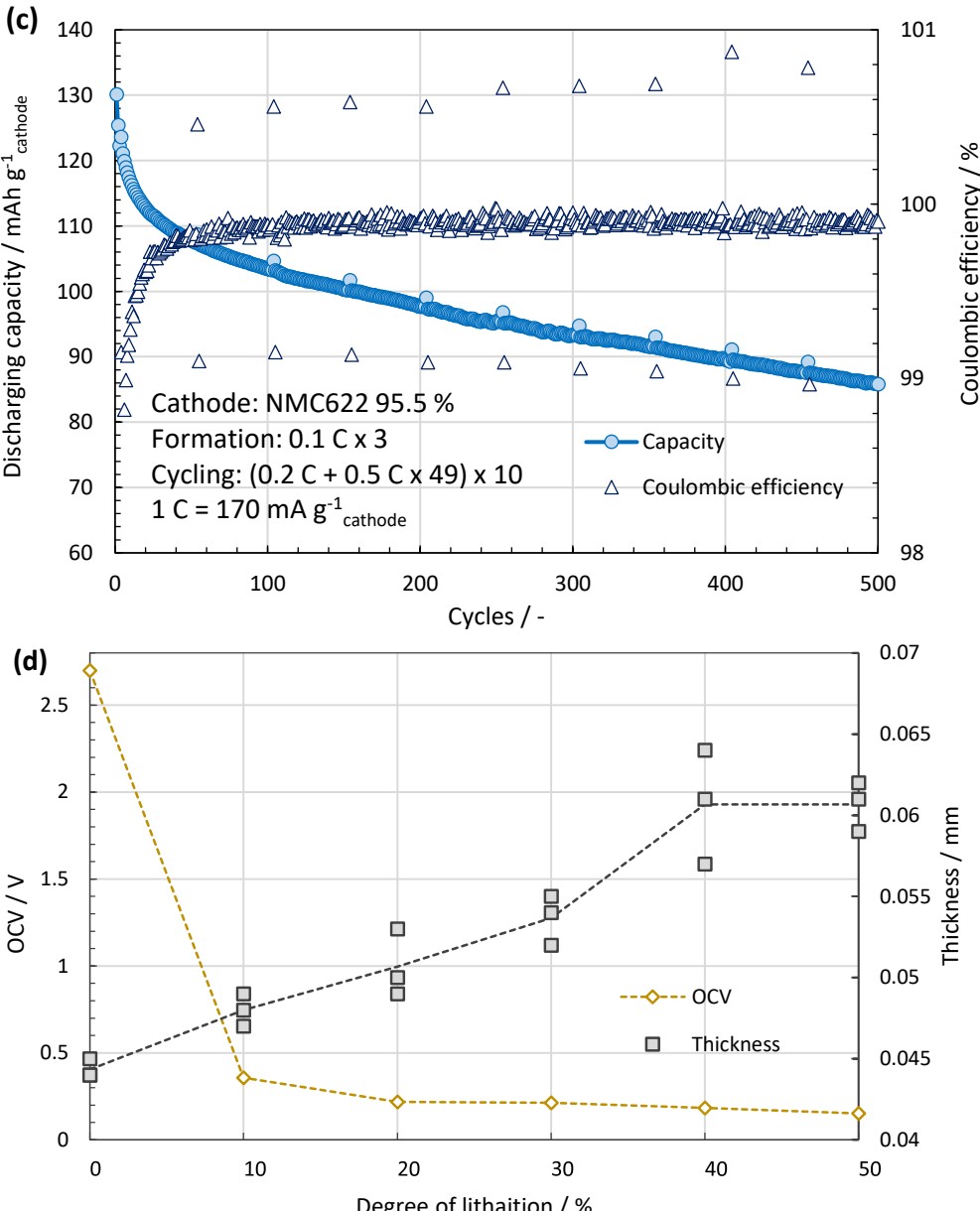

**Figure 3.** (**a**) Discharging profile during lithiation. The process time is controlled according to the pre-defined capacity to be stored in each cell. (**b**) Potential recovery measured for 10 h right after the lithiation process. The OCVs increase at first, but gradually reach a plateau. After this relaxation period, the OCV of the cell is inversely proportional to the degree of lithiation. (**c**) The cyclic behaviour of the made cells shows high Coulombic efficiency and capacity retention. (**d**) With increasing SoC, the thickness of the anode grows from ~44 μm at 0% to ~60 μm at 50% SoC.

Similar coin cells are tested for cycling stability, see Figure 3c,d. After 200 cycles, the capacity retention is 80% of the initial discharging capacity with a constant Coulombic efficiency close to 100%. The anode volume expands linearly towards 50% SoC with approximately 37% increase compared to the pristine anode (from 44 to ~60 μm). These lithiation related volume changes of the anodes are not expected to affect IBA since it is not sensitive to distances.

**Table 2.** Details of the lithiating parameters. All cells charged with the same current of 0.2 C. For comparison, the last column shows the amount of Li as measured by IBA, see below.

| Lithiation Degree/SoC (%) | Anode Capacity (mAh/cm$^2$) | Lithiation Charged (mAh) | Lithiation Charged (10$^{19}$ Atoms/cm$^2$) | Process Time (h) | IBA Detected Li (10$^{19}$ Atoms/cm$^2$) |
|---|---|---|---|---|---|
| 0 | 3.78 | 0 | 0.00 | 0 | 0.057 |
| 10 | 3.78 | 0.38 | 0.85 | 0.5 | 0.892 |
| 20 | 3.78 | 0.76 | 1.70 | 1 | 1.668 |
| 30 | 3.78 | 1.13 | 2.55 | 1.5 | 2.364 |
| 40 | 3.78 | 1.51 | 3.40 | 2 | 3.257 |
| 50 | 3.78 | 1.89 | 4.25 | 2.5 | 4.274 |

## 4. Experimental Setup

After sample preparation, the charged samples are sealed in glass bottles and sent to the post-mortem analysis facility [35]. A few weeks pass between both, hence an equilibration of the lithium profiles is possible. For analysis, 2960 keV protons with ~5 nA ion current and 10 µC doses are applied with 2 mm diameter beam spots positioned in the anode centre, see Figure 4. For the lateral scan along one sample, 0.9 mm diameter spots with 4 µC of ion dose are applied.

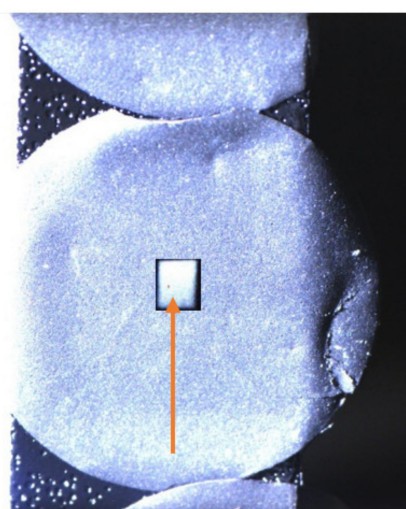

**Figure 4.** Exemplary camera image of the 14 mm circular sample with analysis-spot scintillation-light intensity overlay in the centre, indicating the 2 mm analysis spot size. After IBA, no visible changes of the sample occurred.

The 11 keV FWHM resolution RBS detector is used to obtain the RBS and NRA spectra. The spectra are analysed using SimNRA 7.03 [51] together with the Paneta cross-sections for lithium [45] and RBS or Sigmacalc [52] cross-sections for all other elements. The depth profile analysis employed six layers arranged in a way to cover the full IBA analysis range as outlined in Section 2. Their composition is optimised using MultiSimnra [53]. Here two individual fitting regions are set for the elastic and the inelastic parts of the spectrum. The inelastic part is weighted with a factor of 100 in order to equalize its importance for the $\chi^2$ optimizer to the elastic region, which has intrinsically more counts. The factor 100 is chosen according to the rough ratio of counts per channel in the two regions. Figure 5 shows an exemplary fit following this procedure, resulting in $\chi^2 = 87.4$ for a spectrum with ~3.5 × 10$^6$ counts in the elastic and ~90,000 counts in the inelastic region. The high counting statistics result in negligible statistical measurement uncertainties and relative uncertainties of ~5% between different samples dominated by uncertainties of Particle×Sr, detector

live-time, and fitting quality. Absolute uncertainties relate mostly to the Particle×Sr value derived from the RBS part, and the nuclear reaction cross-section data. These uncertainties sum up to ~7%.

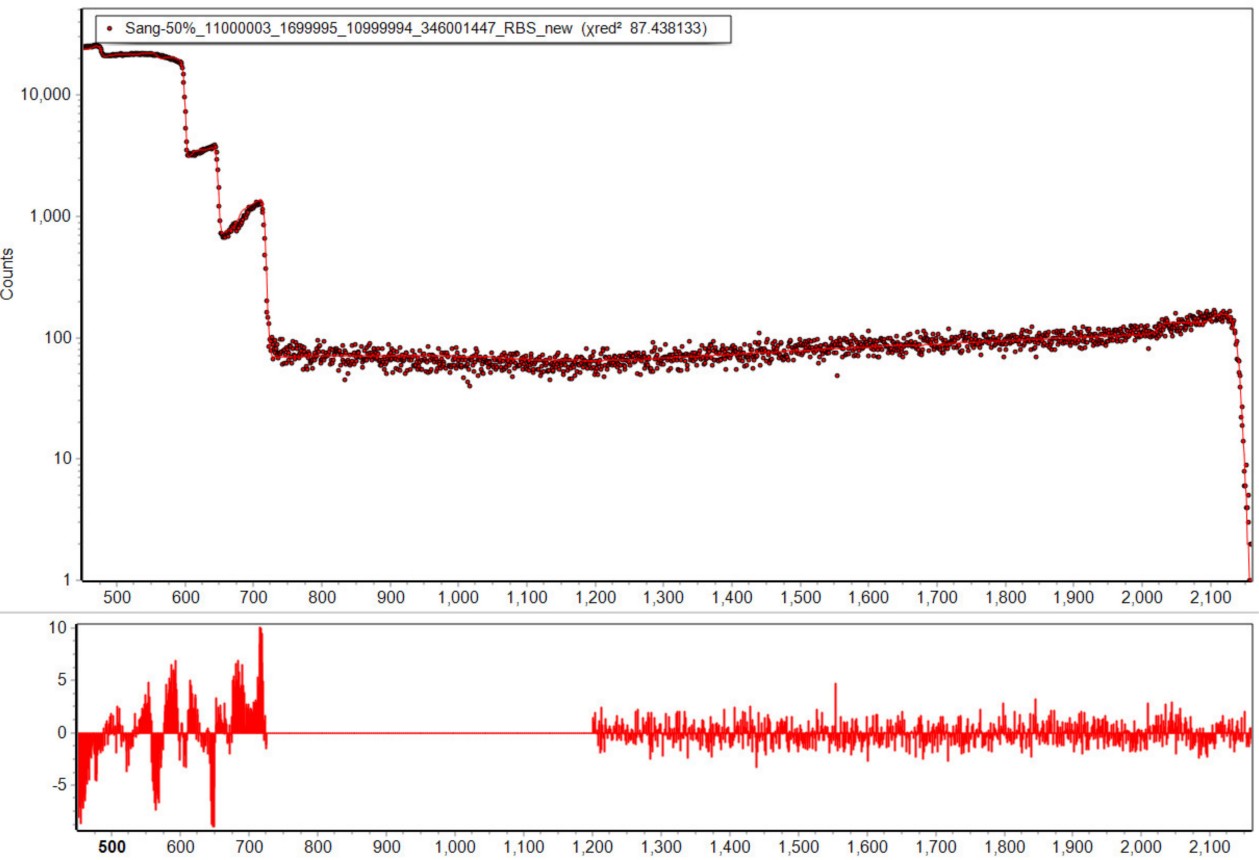

**Figure 5.** Fitted 50% SoC IBA spectrum together with the fit residuals (bottom). The fitting by Multisimnra yields a good agreement and confidence in the resulting depth profiles.

## 5. Results

Figure 6 demonstrates the depth profiles obtained from this fit of the 50% SoC sample (a) and the comparison of the Li concentration between all six samples (b). The sample thickness is evaluated by assuming a stack of six layers with increasing thickness towards the sample backside. We see 7.7 at.% of Si, 7.6% O, 14% Li, and the remaining 70.7% C in this sample. With the lower Li content in the other samples, the other elements increase their share with constant ratios within uncertainties. The 0% SoC sample features 8.6 at.% Si, 8.6% O, 0.2% Li, and the remaining 82.6% C. The detection of H is not directly possible with the applied method, but it can be detected indirectly as a missing element for a consistent sample composition. Since RBS, NRA, and PIXE can actively detect practically all other elements such as N, C, O, or Si, it is highly probable that the missing aspect can be accounted to H. Nevertheless, this indirect nature significantly increases the uncertainty of H concentrations shown in Figure 6, in comparison to the other elements. The measurements show a few % H in the samples with most of it sitting near the surface in the reaction layer.

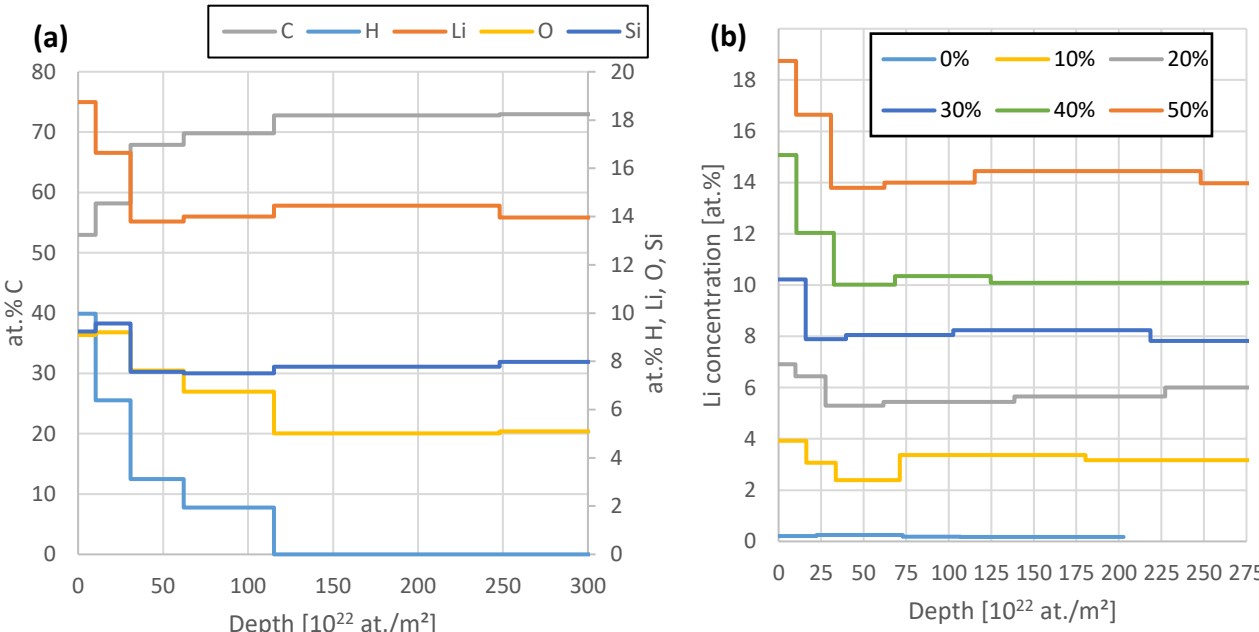

**Figure 6.** (**a**): Comparison of the elemental depth profiles of the 50% sample. All elements except C are depicted on the right axis. After a surface reaction layer of ~3 μm thickness, all elemental profiles are constant within uncertainties. (**b**): Comparison of the Li depth profiles of all samples. The region >200 × $10^{22}$ at./m$^2$ has to be considered with care due to the dominance of the pile-up signal in this region.

In all cases, except for the 10% case, the Li concentration remains mostly constant, besides a thin surface reaction layer. The 10% case shows a slight dip in the Li concentration of 2 at.% starting after the surface reaction layer, with ~3.2% in the remaining regions. It remains unclear whether the increased Li concentration close to the surface relates to the intercalation process, since the Li infiltrates the anode from this side and a maximum concentration could be expected here, or if it relates to the air exposure, since also H and O are higher in this region, or the formation of a solid-electrolyte interface (SEI). Figure 7 compares the Li concentration average excluding the first 40 × $10^{22}$ at./m$^2$ and the ratio of maximum Li concentration to this average. Table 2 compares the injected amount of Li derived from the charged capacity and the total amount of Li found by IBA. Interestingly, we find a mostly constant ratio of maximum to mean Li concentration of 1.23. The Li depth profile of the 0% SoC sample is constant within the somewhat larger uncertainties with no pronounced surface peak/maximum. Equation (1) depicts the resulting linear fit of Figure 7a with R$^2$ > 0.999.

$$\text{Li [at. \%]} = 0.2 + 0.29 \times \text{SoC[\%]} \qquad (1)$$

The Si depth profiles are shown in Figure 8. Similar profiles are obtained for O. The results indicate a 20% enrichment of SiO$_x$ at the surface (up to 30 × $10^{22}$ at./m$^2$ depth), although the results somewhat scatter due to inaccuracies in the data fitting in this region and the mass resolution of the given RBS setup cannot distinguish between Si and Al enrichment. The O concentration is also higher near the separator side.

The lateral homogeneity of the 50% sample is checked using a radial line scan of 10 points with ~0.8 mm step length. The lateral analysis yields no significant variations of the layer composition within uncertainties. The observed copper signal suggests slight variations in the anode layer thickness, but the roughness and the thickness of the anode layer being close to the IBA range make quantification impossible.

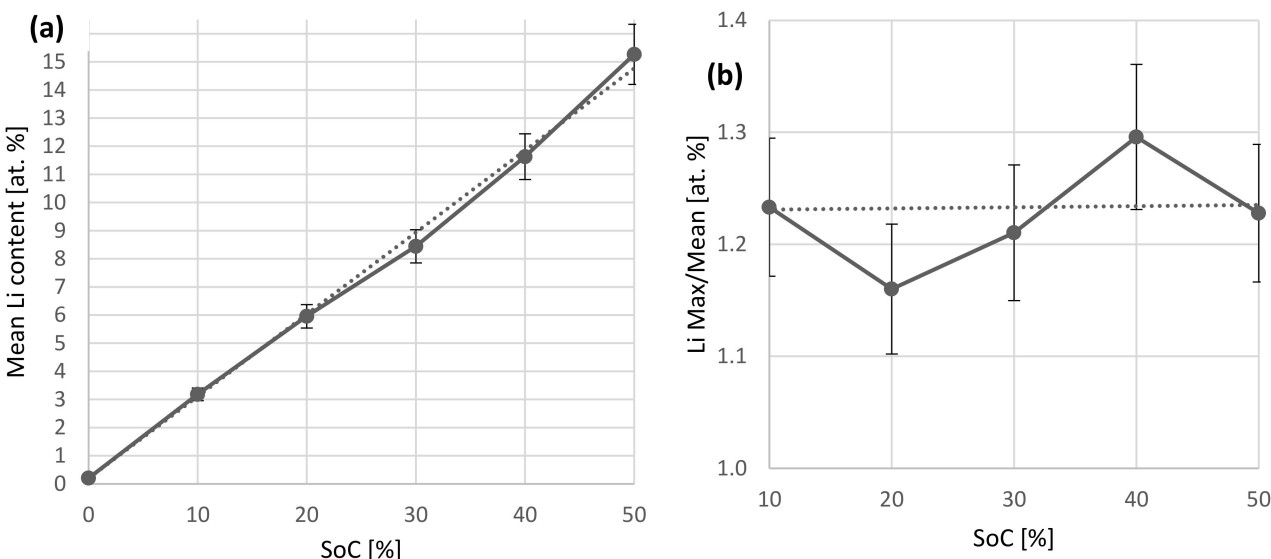

**Figure 7.** (**a**): Plot of SoC against the average lithium concentration. The data agree with the expected linear connection between these parameters. An offset of 0.2 at.% Li is found in the as-prepared state originating from the LiPAA binder. (**b**) Plot of the surface near Li concentration divided by the average (**a**). The ratio is constant within relative uncertainties.

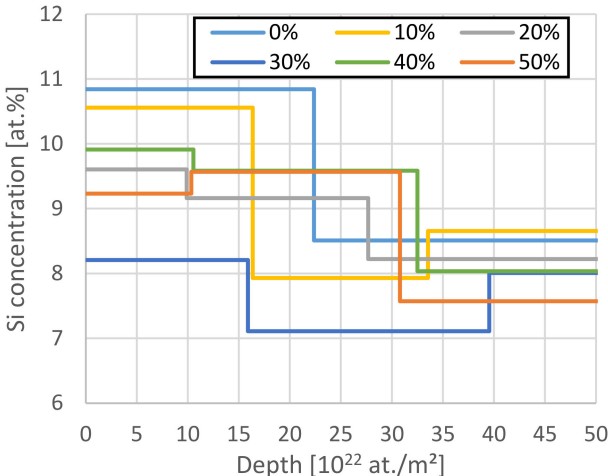

**Figure 8.** The Si depth profiles measured by RBS. The data evaluation for Si is limited to $50 \times 10^{22}$ at./m$^2$ due to the overlap with the O RBS signal. The data show a slight increase by ~20% of the Si concentration near the surface, but due to the close masses, this apparent Si-enrichment could also indicate a surface-near deposition of Al.

Figure 9 shows representative SEM pictures of the separator side sample surface after lithiation. The anodes feature significant porosity and consist of SiO$_x$ and C particles. A negligible amount of sub-μm-sized Si-containing fibres is found, but these originate from the applied fibre-glass separator. A histogram analysis of the amount of pixels with light and dark grey levels yields about 15% surface coverage of the SiO$_x$ particles and 80% of graphite, with the remainder related to dark areas of the porosity. The accuracy of this approach is limited, due to the high porosity and the grey-scales it introduces into the image. The result agrees quite well with the educt mixture, but it cannot support or disprove the observed ~20% Si surface enrichment. Figure 10 extends this surface investigation using a focussed-ion-beam (FIB) cut through the anode, and Figure 11 shows a cross-cut. The porosity extends through the anode depth. The given FIB cross-sectional area only features

11 SiO$_x$ grains, hardly allowing for any statistically sound profile information. The cracked surface features more statistics, but the grey-scale and EDX analysis show no clear depth profile or trend. The natural roughness of a cracked surface limits the accuracy of this approach, though. The EDX analysis shows a surface-near (few µm) presence of Al, but the sample structure prevents a quantification. This supports the assumption of an Al signal overlap with the Si signal in the above RBS analysis, but due to the missing quantification of the EDX data it remains unclear if the quantity of Al is sufficient to explain the apparent enrichment of Si near the separator interface.

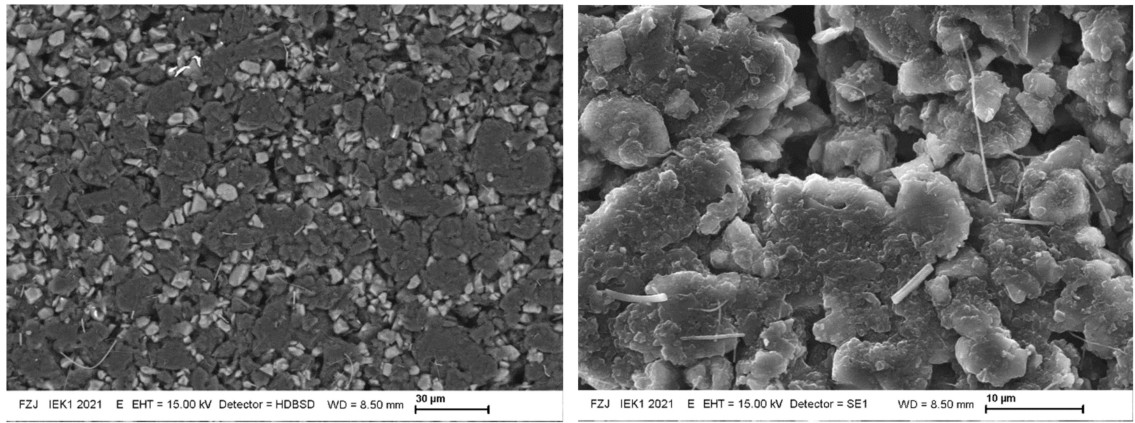

**Figure 9.** 15 kV SEM pictures of the top surface of the 30% SoC sample. The electron range is <3 µm in the material. EDX confirms the light grey areas to be Si/SiO$_x$ rich, while the dark grey regions consist of C. The Si-rich fibres have <1 µm diameter and apparently exist besides the more spherically-shaped Si-rich particulates of 1–10 µm diameter on the surface.

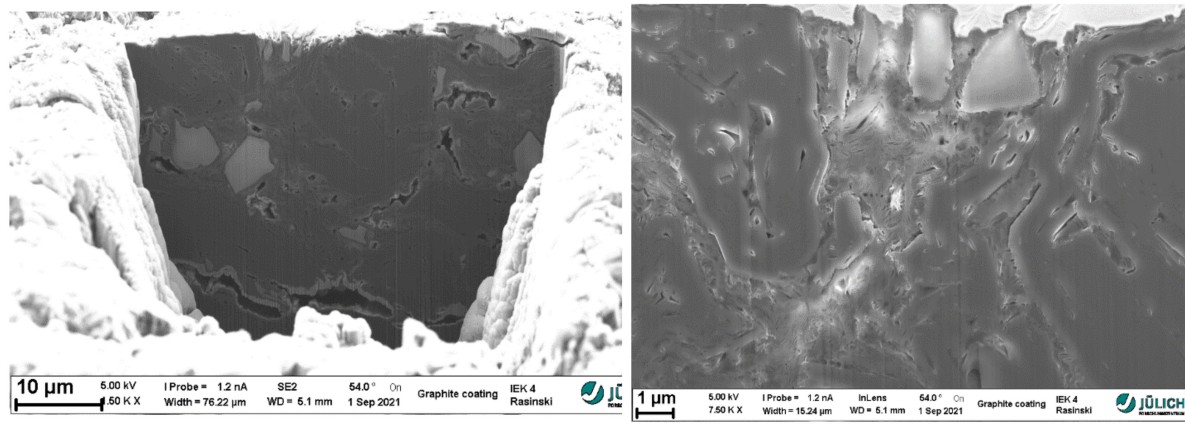

**Figure 10.** Five kilovolt SEM pictures of a 33.5-µm-deep FIB cut in the separator side surface of the 50% SoC sample. SiO$_x$ particles with sizes between 1–3 µm and substantial porosity are visible, but there is no clear indication of thicker SEIs. An EDX scan through the four brighter particles on top of the right figure confirms them to be SiO$_x$ particles.

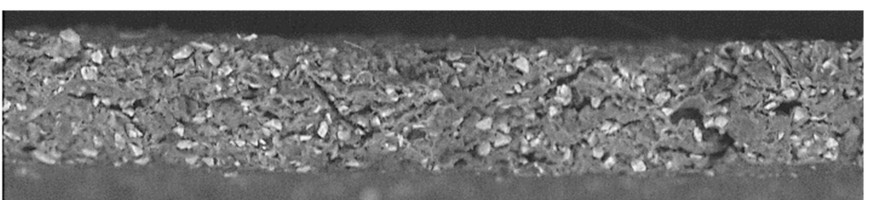

**Figure 11.** Fifteen kilovolt backscattered electron SEM image of a cross-cut of the 40% SoC anode with the separator side on top. The light grey particles are SiO$_x$ and the dark grey C. The investigated anode region is $400 \times 55$ µm$^2$.

## 6. Conclusions

### 6.1. IBA Method

The study used MeV ion-beam analysis (IBA) for investigating the lithiation depth and lateral profiles of a SiOx-containing graphite active material lithiated in a coin cell. The problems observed in mixed cathode and electrolyte analysis of significant immobile lithium, which deteriorates the accuracy for the mobile lithium fraction [40], are not relevant for the analysis of anodes/anode materials presented here. The low immobile Li background of 0.2% is negligible for the overall signal beyond a few 1% SoC. The lithiation degree was successfully determined up to the total sample thickness of about 44 μm. The IBA data for Li concentrations are proportional to the electro-chemically injected charge (Ah) at all SoC (Table 2). All samples showed a reduction of Li concentration with depth at the separator side, proofing the depth-resolved analysis of the lithiation degree using the selected $^7Li(p,\alpha_0)^4He$ reaction. Calculations of the irradiation-induced changes show the possibility to apply orders of magnitude higher ion doses and currents without relevant changes of the sample structure or temperature, confirming the non-destructive nature of IBA as supported by the visual inspection after analysis. Only for situations requiring μm-sized spots, the local heat-load could become a critical factor. The good counting statistics observed in our data leave room for reducing ion current and doses <<10 μC without a significant impact on the total uncertainty of 5%, to further mitigate such problems.

In conclusion, our results indicate no fundamental limitation of the application of IBA to lithium and lithium-ion batteries. Besides the Li-based examples shown here, IBA can be used to analyse any cell type and component such as sodium-based cells, metal anodes, liquids, solids, and powders without major methodical changes. The analysis range is limited due to fundamental constraints of the nuclear reactions and the product detection. For $^7Li(p,\alpha_0)^4He$, we obtained 46 μm range for Li in the carbon-based anode material, which is supported by calculations and finding identical amounts of Li ions in the anodes through lithiation (mAh/cm$^2$) and IBA. The calculations show a perspective for even higher ranges when using deuterons or IBA methods other than NRA. The missing cross-section data should be determined for those options as soon as possible. In the future, the experiments will be extended towards an in-situ analysis of the lithiation depth profiles during (dis-) charging for revealing the lithium kinetics and the migration of other elements during cell operation. The current data suggest the possibility for a time resolution of 5–10 min (up to 12 C rate) for a relative accuracy of ~2%.

### 6.2. Si-Doped Anodes for Li Batteries

The amount of injected charge recalculated from mAh to Li-ions shows a good agreement with the amount of Li atoms found in the samples by IBA. Combining the knowledge of maximum loading of Si and C with the determined Si and C quantities by IBA allows for determining the anodes SoC with a method independent of the electro-chemistry. The analysis revealed a C/Li = 5.4 (when neglecting Si) and a Li/Si = 1.8 (when neglecting C) at 50% SoC (compared to Li/Si $\leq$ 3.75 for pure Si). IBA cannot disentangle whether the Li preferably binds to Si/SiO$_x$ or C, but since already at 50% SoC the C/Li ratio is lower than the typical limit of C/Li = 6 (LiC$_6$), the SiO$_x$ definitely binds to at least a share of the injected Li at the highest investigated lithiation degree, as would be expected from electrochemistry. The presence of inactive C from the binder is not even considered here. We can further investigate this question by adding information from the electro-chemical analysis. Extrapolating the Li loading ratios with the given C/Li = 6 limit and the electro-chemically determined capacity of 622 mAh/g, we obtain a limit for Li storage in the used SiO$_x$ of Li/Si = 1.04 $\pm$ 0.05. This value of Li uptake in the used SiO$_x$ of ~1:1 is about four times lower compared to the value of Li/Si= 3.75 for pure Si, but still represents a 67% increase in specific capacity of the investigated C/Si$\approx$9 (IBA elemental ratio) material compared to pure graphite (372 mAh/g).

The depth analysis revealed a constant Li enrichment factor at the anode–separator interface. The gradient consistently shows a factor of 1.23 when comparing the surface-near

maximum with the in-depth average, independent of the SoC (up to the investigated 50% SoC). The 0% SoC sample (this is not installed into a cell) features a constant Li depth distribution, indicating that the surface-near maximum in the other samples relates to their installation into the coin-cell and/or the charging process. The gradient extends from the separator interface to ~6 μm depth in the anode, a depth corresponding to the centroid of the $SiO_x$ particle size distribution. Since the samples are exposed to air for installation into the IBA device, this enriched region could also originate from air reactions at this side. We would expect a constant fraction of Li in this case, related to the presence of e.g., a LiOH layer, not the observed SoC-dependent concentration. The formation of a solid-electrolyte-interface (SEI) also cannot explain this effect. Assuming a constant porosity through depth, the SEI present on the internal surfaces would add a constant contribution to the Li depth profile, not only a surface near contribution, since it grows equally throughout the anode depth. Another possible explanation is a higher Si concentration near the surface, in which the Li would bind preferably. This could originate from the different particle sizes of graphite and $SiO_x$ (factor ~2.6), which is known to induce particle size segregation, but the SEM data are inconclusive on this particle depth distribution due to the limited particle distribution counting accuracy. The preferable binding of Li to Si compared to C is already indicated by the C/Li ratio discussed above, supporting a connection between Si and Li. The data on the Si depth profiles show a significant scatter, but within, uncertainties on average-matching Li and $SiO_x$ enrichment profiles are found at the separator interface. Unfortunately, due to the limited atomic mass resolution of the IBA setup, the measurements remain inconclusive if the Si enrichment near the separator side exists or if there is an Al deposition on the anode as the EDX analysis suggests. This Al, probably originating from the fibre-glass separator as the presence of sub-micron glass-fibre fragments on the anode surface suggests, could bind additional Li or even show a particular beneficial behaviour in combination with $Si/SiO_x$. Lastly, the gradient could originate from the diffusion of Li into the anode with a depth-dependent concentration given by the lithiation rate and temperature, but further measurements with different conditions would be required to confirm this option. Consequently, at least one modification of the anode composition in terms of Al or $SiO_x$ concentration induced inhomogeneities in the Li depth profiles and potentially the local SoC.

**Author Contributions:** Conceptualization, S.M., E.F. and M.F.; data curation, S.M. and H.J.; formal analysis, H.J.; investigation, S.M., H.J., M.R. and M.M.; methodology, H.J.; resources, M.F.; software, S.M.; supervision, E.F. and M.F.; visualization, S.M.; writing—original draft, S.M.; writing—review & editing, H.J., E.F. and M.F. All authors have read and agreed to the published version of the manuscript.

**Funding:** This work was funded by the German Federal Ministry of Education and Re-search (BMBF) within the "InCa" project (FKZ: 03XP0228D) and the "Meet Hi-EnD III" project (FKZ: 13XP0258B) and by the Deutsche Forschungsgemeinschaft (DFG, German Research Foundation) – 491111487, which is gratefully acknowledged here.

**Institutional Review Board Statement:** Not applicable.

**Informed Consent Statement:** Not applicable.

**Data Availability Statement:** Data are available upon request from the corresponding author.

**Conflicts of Interest:** The authors declare no conflict of interest.

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
