# Peer review of "Quantitative Lithiation Depth Profiling in Silicon Containing Anodes Investigated by Ion Beam Analysis"

_batteries, doi:10.3390/batteries8020014_

Round 1

Reviewer 1 Report

In this work, a specific type of lithium ion battery anodes, Si-doped anodes, was chosen to be the study subject of lithiation process. With the controlled doping ratio, the cross-section SEM and EDX images showed clearly the reaction between the anode material and lithium ions. The data was presented in a logic way and was further analyzed the IBA method. This work is a very meaningful step of mechanism study on SEI and anode materials.

I think this manuscript can be accepted after minor revision. Some error messages on citation generated by the software need to be taken care of. 

Author Response

Dear Reviewer,

thanks for your positive comments. We will check the citations and fix the problems.

Reviewer 2 Report

This paper reports on a comprehensive characterization of Si anodes by means of ion beam analysis. This is suitable for publication in Batteries after the authors address these comments.

  1. Why higher Li/Si concentrations in the ~6um surface interphase indicates Li binding to SiOx?
  2. Fig 3. Why is the specific capacity of Si 622 mAh/g? Generally it’s thought to be >3000 mAh/g.
  3. Fig 8. Why is the Si concentration data square wave like?
  4. Fig 10. Why are there some SiOx particles embedded in the sample? Impurity of starting Si material?
  5. As the authors correctly pointed out, this method could be used to study electrode materials beyond Si electrodes. Can we use it to characterize metal anodes (DOI: 10.1126/sciadv.abb1122; 10.1016/j.matt.2021.09.025) or other intercalation electrodes like graphite (DOI: 10.1038/nature14340; 10.1038/s41560-021-00797-7 )? The authors could discuss the potential application of the method in these fields that are of enormous interest to bring out the broader impact of the paper.

Minor:

  1. spelling of localization/localization should be consistent. E.g. line 13 and line 16.
  2. line 375: “Error! Reference source not found..”

Author Response

  1. Why higher Li/Si concentrations in the ~6um surface interphase indicates Li binding to SiOx?

because of the ratios of the elements. The Si and Li depth profiles both have a maximum near the surface. The concentration of C is lower near the surface, since Si and Li are more and all elements add to 100%. This means Li follows the trend of the Si concentration not the C concentration. The statement was weakened, since this is only an indication, not a proof of the preferential binding. Furthermore C/Li is already smaller than 6, the natural limit for Li intercalation in graphite, in this region so the Si has to contribute to the Li storage.

  1. Fig 3. Why is the specific capacity of Si 622 mAh/g? Generally it’s thought to be >3000 mAh/g.

The specific capacity of the used SiOx+Graphite anode is 622mAh/g. In any case SiOx has a lower specific capacity than pure Si and furthermore its only a fraction of the total anode mass, limiting its impact on the total .

  1. Fig 8. Why is the Si concentration data square wave like?

This is due to the discrete layer-wise nature of the data evaluation software. It can only assume independent layers to fit the IBA spectra.

  1. Fig 10. Why are there some SiOx particles embedded in the sample? Impurity of starting Si material?

We only used SiOx, no pure Si. Maybe this was not 100% clear from the text. Consequently we changed the description in abstract and introduction to make this more clear.

  1. As the authors correctly pointed out, this method could be used to study electrode materials beyond Si electrodes. Can we use it to characterize metal anodes (DOI: 10.1126/sciadv.abb1122; 10.1016/j.matt.2021.09.025) or other intercalation electrodes like graphite (DOI: 10.1038/nature14340; 10.1038/s41560-021-00797-7 )? The authors could discuss the potential application of the method in these fields that are of enormous interest to bring out the broader impact of the paper.

hm, thanks for the hint. The analysis of other elements/electrode materials shouldnt be a problem, although the analytical capabilities in terms of range and depth resolution could be different from what was shown here. In fact a project for the analysis of Li in pure Al anodes is ongoing.

Statements added to introduction and conclusions.

Minor:

  1. spelling of localization/localization should be consistent. E.g. line 13 and line 16.

Corrected

  1. line 375: “Error! Reference source not found..”

we did not find this in our version, maybe a software problem. We will keep an eye open in the proof.

Reviewer 3 Report

Reviewer

General comment: This manuscript reported that “Quantitative lithiation depth profiling in silicon-containing anodes investigated by ion beam analysis”. The methodical developments of MeV ion beam analysis (IBA) presented here open up new possibilities for simultaneous elemental quantification and localization of light and heavy elements. It describes in detail the technical prerequisites and limitations of using IBA to analyze and solve current challenges in Li-ion and solid-state battery-related research and development, which is quite interesting. Therefore, I suggest this manuscript can be accepted after minor revisions.

Comment 1: The authors should improve their literature review by adding more recent references

Comment 2: It is better to add the cycling performance and columbic efficiency of the coin cell. Comment 3: The authors should discuss the volume expansion of anode materials after cycling.

Comment 4. The conclusion should be more concise and clear.

Author Response

Dear Reviewer 3,

thanks for your comments.

Comment 1: The authors should improve their literature review by adding more recent references

we are open to add recent literature. references were added in view of other reviewer comments. Please state specific fields or paper you are thinking of in addition if this is not sufficient.

Comment 2: It is better to add the cycling performance and columbic efficiency of the coin cell.

Numbers and a short explanation were added in section 3.2. We have graphs/data, but since there are no real surprises there we would like to omit the graphs in order to keep the paper compact. We can add them to the digital appendix.

Comment 3: The authors should discuss the volume expansion of anode materials after cycling.

A statement was added in the sample preparation section see comment 2. A statement on the effect of volume changes for IBA was added.

Comment 4. The conclusion should be more concise and clear.

We checked the conclusion and improved the clarity.

Round 2

Reviewer 3 Report

Comment 2: It is better to add the cycling performance and columbic efficiency of the coin cell

I think Author’s reply is not good enough. It is a very important data for Battery; therefore, it would be better to add this data in the main manuscript (as Figure 3c and d).

Author Response

Dear reviewer, the two graphs were added as requested.

Round 3

Reviewer 3 Report

Accepted

Author Response

I didnt see any comments anymore, so thanks for the review